# Neuroimaging findings and balance problems after mild traumatic brain injury: A systematic review protocol

Todd Boscarello[1]*, Robby Boparai[1], Nathan Samson[2], Alan Rodriguez[2,3], Thomas Knoblauch[2,3], Cheryl Vanier[2,3,4], Travis Snyder[2,3]

**1** Reno School of Medicine, University of Nevada, Reno, NV, United States of America, **2** Touro University of Nevada, College of Osteopathic Medicine, Henderson, NV, United States of America, **3** IMGEN Research Group, Las Vegas, NV, United States of America, **4** Touro University Nevada: A JBI Affiliated Group, Henderson, NV, United States of America

* tboscarello@med.unr.edu

## Abstract

### Objective

To systematically review studies relating neuroimaging findings to balance problems resulting from a history of mTBI.

### Introduction

Mild traumatic brain injury affects 55.9 million people worldwide every year. These injuries can have persistent symptoms such as maintaining balance which can be life-altering. Difficulties maintaining balance persist months or years after a mild traumatic brain injury in >30% of patients. Neuroimaging modalities, including magnetic resonance imaging, diffusion-weighted imaging, functional magnetic resonance imaging, electroencephalography, and magnetoencephalography, have been associated with presentation or persistence of balance difficulties, but no clinical guidelines are currently in place.

### Inclusion criteria

Studies will include participants of any age or sex who were diagnosed as having mild traumatic brain injury by a medical professional, excluding studies which by design included patients with other conditions diagnosed using neuroimaging findings. There must be at least one post-injury scan from at one or more of the included neuroimaging modalities, and assessment of balance problems. A comparator must be present in the form of either a control group or longitudinal design.

### Methods

A search will be conducted in Elsevier (Embase), MEDLINE (PubMed), Google Scholar, SportDiscus (EBSCOhost) and ProQuest for studies meeting the inclusion criteria, published 2013–2024, and available in English. Reviews will not be included. The process of study selection, critical assessment, data extraction, and summarizing findings will be

**Data Availability Statement:** No datasets were generated or analysed during the current study. All relevant data from this study will be made available upon study completion.

**Funding:** The author(s) received no specific funding for this work.

**Competing interests:** The authors have declared that no competing interests exist.

conducted by two independent reviewers, with disagreements resolved by a third. The meta-analysis will summarize the strength of association between specific findings related to brain regions using various neuroimaging modalities and the presentation or persistence of balance difficulties. Evidence related to each neuroimaging modality will summarized using the GRADE approach.

## Trial registration

**Systematic review registration number: CRD42024476988**.

## Introduction

Mild traumatic brain injury (mTBI), sometimes called concussion, is a disruption of brain function resulting from trauma. An mTBI can be distinguished from moderate or severe TBI based on duration of loss of consciousness (LOC; must be <30 minutes for mTBI), the Glasgow Coma Scale (GCS) 30 minutes after the injury (mTBI has GCS between 13 and 15), and duration of post-traumatic amnesia (mTBI is indicated if <24 hours) [1]. Signs in the acute phase (<72 hours after injury) may include changes in mental status (e.g., confusion, disorientation), physical functioning (e.g., headache, balance problems, vision problems), cognitive symptoms (e.g., memory problems, difficulty concentrating), and changes in experience of emotion (e.g., unusual emotional lability or irritability) [1]. Between 30% and 80% of patients diagnosed with mTBi have symptoms that persist months or years past the acute phase [2].

The diagnosis and prognosis of patients with persistent symptoms has been the subject of vague and changing clinical guidelines. For example, the term 'Post-Concussive Syndrome' (PCS) is common, but the diagnosis of PCS has been controversial. It is not in the DSM-5 [3], but is in the ICD-10 [4]. The ICD-10 PCS diagnosis requires three of the following symptoms to persist ≥3 months after a mTBI: headache, dizziness, fatigue, irritability, insomnia, concentration or memory difficulty, or intolerance of stress, emotion, or alcohol. Dizziness is strongly associated with loss of balance, and this symptom alone can reduce quality of life for patients after mTBI.

Neuroimaging can play an important supporting role in prognosing patients with brain injuries, with different neuroimaging modalities targeting distinct characteristics of the brain. Conventional magnetic resonance imaging (MRI), diffusion weighted imaging (DWI), functional MRI (fMRI), electroencephalograms (EEGs), and magnetoencephalograms (MEGs) may associate with persistent symptoms after mTBI [5]. Conventional MRI studies suggest brain volume changes in both cortical regions and subcortical structures occur over time after mTBI [6,7]. Conventional MRI can also identify white matter hyperintensities, which have been associated with age and neurodegenerative diseases, but with mixed evidence for association with mTBI symptoms [8]. A review of conventional MRI findings after mTBI found that approximately 30% of conventional magnetic resonance imaging (MRI) studies found significant relationships with the burden of gray or white matter lesions, volumetric changes, changes to blood flow, or other signs of pathology [5]. The brain regions associated with chronic mTBI symptoms included volume of cingulate gyrus isthmus, ventromedial prefrontal cortex, and fusiform gyrus [5].

DWI and its most frequently encountered variant, diffusion tensor imaging (DTI), use MRI to analyze water diffusion in tracts within the brain to reveal microscopic details of tissue structure that relate to white matter connectivity. Several standard metrics emerge from a DTI

study, including mean diffusivity (MD, which reflects the total movement of water), fractional anisotropy (FA, which indicates the degree of constraint associated with the direction of water movement), radial diffusivity (RD, which indicates diffusion perpendicular to axon fibers), and axial diffusivity (AD, which indicates diffusion parallel to axon fibers) [9]. A scoping review found that 54% of studies reported associations between PCS and either fractional anisotropy (FA) or mean diffusivity (MD) measures in brain regions such as the corpus callosum, longitudinal fasciculus, and tracts of the internal capsule [5]. A systematic review of the relationship between DTI and development or severity of PCS found that smaller FA and higher MD and RD were associated with PCS, particularly in the corpus callosum [10].

Functional MRI analyzes blood flow to indicate activation during task and resting state functional connectivity. The fMRI findings were most promising in associating with PCS: a scoping review found that 83% of fMRI studies showed some association with PCS [5]. Activation of various cortical regions and subcortical structures, such as the parahippocampal gyrus, posterior cingulate gyrus, thalamus, and hypothalamus, have been associated with PCS [11,12].

An EEG provides information about electrical activity in the brain, whereas a MEG measures the brain's magnetic fields. Both electrical and magnetic fields are generated from currents associated with the movement of ions in neurons, and they are considered a measure of functional connectivity [13]. In the scoping review, 78% of M/EEG studies showed some association with PCS [5].

Given the disparate etiology of the symptoms used to diagnose PCS, it is unsurprising that, although associations between neuroimaging and PCS have been reported, there is high variability in findings and no reliable neuroimaging findings have emerged for chronic symptoms after mTBI [5,14]. Better indicators may improve patient outcomes by suggesting which patients need early interventions for persistent sequelae of mTBI. The potential impact could be substantial: mTBI affects an estimated 55.9 million people each year worldwide [15], and 16.8-44.7 million will develop persistent symptoms [2]. Better diagnostic criteria would also help differentiate chronic mTBI from neurodegenerative diseases, which require different treatment approaches.

Narrowing the focus from the broad PCS symptom criteria to a single symptom may help in the search for clinically useful neuroimaging findings. Apart from the use of dizziness in diagnosing PCS, balance problems or dizziness are a relatively understudied symptom associated with chronic mTBI, even though it is estimated that up to 32.5% of mTBI patients may experience dizziness two years or more after the injury [16]. Balance problems may or may not significantly affect quality of life in younger mTBI patients, but factors that increase risk of falling can greatly impact older patients. A fall in an older patient can lead to fractures, hospitalization, and loss of independence, and it can start a decline which hastens death. Therefore, this systematic review will focus on balance problems. A recent review identified 234 brain regions which were implicated in influencing balance, with the strongest evidence of association with the cerebellum, basal ganglia, thalamus, and hippocampus [17]. This gives us reason to believe that the various neuroimaging modalities will reveal pathology in a subset of the identified brain regions in mTBI patients.

A preliminary search of MEDLINE, Google Scholar, the Cochrane Database, and JBI Evidence synthesis yielded no reviews published within the past ten years or underway that covered the same topic. There are existing reviews (e.g., [5]) associated with neuroimaging and PCS or elements thereof. Dizziness was not individually studied. It is also clear from the reviews of other symptoms that some brain regions and neuroimaging modalities provide unique information about persistence of one type of symptom and not others (e.g., [18]). Another review linked issues with maintaining balance to brain regions, but not in the context

of mTBI [17]. The objective of this review is to inform clinicians and guide future research regarding which neuroimaging modality or modalities is/are most informative for the presentation or persistence of balance problems in mTBI patients. Although the main clinical focus is on persistence of balance symptoms, presentation is included to identify continuity in acute and chronic neuroimaging findings. For example, changes in blood flow and directions of water diffusion, apparent in fMRI and DTI soon after an injury, may lead to atrophy that is detectable in volumetric MRI weeks later. Many studies of neuroimaging as prognostic factors have a single brain scan with longitudinal follow-up of symptoms.

### Review question

Based on the Population, Index model, Comparator model, Outcome(s), Timing, and Setting of the study, the review question is: Which neuroimaging modality yields findings that are reliably associated with presentation and persistence of balance problems after mTBI? Secondarily, which brain regions are most reliably associated with presentation or persistence of balance problems after mTBI?

## Methods

This protocol has been written according to PRISMA-P guidelines for the systematic review and meta-analysis of scientific research [19], the JBI methodology for systematic reviews for etiology and risk [20], and guided by recommendations for prognostic factor systematic reviews [21]. It is registered with PROSPERO (CRD42024476988).

### Inclusion criteria

**Population.** The target population in the review is patients diagnosed with an mTBI by a medical professional. Participants of any age will be included in the study, although age subgroups, particularly delineating those under 18 from adults, with an additional sub-group for older adults, will be recorded separately for subgroup analysis when available.

Studies will be excluded if they include by design participants with conditions that are typically diagnosed through neuroimaging, such as neurodegenerative disorders, stroke, transient ischemic attack, tumors, venous or arterial problems, infections such as Creutzfeldt-Jakob disease, encephalopathy, or inflammatory conditions such as multiple sclerosis [22]. Incidental inclusion of participants with such conditions will not be grounds for exclusion.

**Index prognostic factor: Neuroimaging.** At least one post-injury neuroimaging scan must have been done after the injury leading to the mTBI diagnosis to be included in this review. Brain regions or tracts will be identified as stated in the publication, and the authors will aggregate selected regions according to the NeuroQuant Atlas based on their prevalence in included studies. The prognostic modalities included in this review will be MRI, DWI/DTI, fMRI, EEG, and MEG. Quantifiable metrics for each modality include the following:

Conventional MRI: volumetrics ($cm^3$), lesion load or volume ($cm^3$)

DWI/DTI: fractional anisotropy, axial diffusivity ($mm^2/s$), radial diffusivity ($mm^2/s$), mean diffusivity ($mm^2/s$), diffusion kurtosis [9]

fMRI: functional activation during task, resting state connectivity (T scores) [11])EEG: event-related potential (ERP) component amplitude (microvolts) [23]

MEG: slow wave (femtoteslas or picoteslas) [24]

**Outcomes.** Balance is defined as the ability to control one's center of gravity and body position to avoid falling. Deficits in maintaining balance can be described in a variety of ways, including dizziness, vertigo, or maintenance of postural stability. This review will include studies that describe balance problems as a general balance or risk of falling issue, 'dizziness',

'vertigo', or 'postural stability'.' It can be assessed using a standard tool e.g., [25,26] or self-reported, since there is high concordance between self-reported balance problems and those assessed using standard tools [27]. Studies included in the review must have participants with diagnosed mTBI and balance problems, with information on the presence of balance deficits and/or how long the balance problems have persisted since injury.

**Timing.** At least one neuroimaging scan must have taken place within five years from date of injury. Timing of the scan, particularly whether it was done during the acute/subacute phase (24 hours to 7 days/1 week to 3 months, respectively) or chronic phase (3 or more months) after mTBI is an important consideration, as there may be an interaction between prognostic accuracy of a neuroimaging modality and timing of the scan. Longitudinal studies, in which participants were scanned twice or more will also be included. Time to resolution of symptoms of balance symptoms will be recorded when available. To maintain consistency with criteria for PCS, balance symptoms will be considered long-term if they persist three months or more post-injury.

**Setting.** The prognostic value of neuroimaging modalities for managing and treating balance problems is relevant to primary and secondary care settings after mTBI.

**Types of studies.** The review will include any type of primary study reporting original data, such as case studies (where-in data fits our criteria), case series, cohort studies, case control studies, cross-sectional studies, quasi-experimental, or experimental studies. Reviews and commentaries will be excluded.

## Search strategy

A systematic search will be conducted using trusted sources to identify relevant studies based on the criteria for population, prognostic neuroimaging techniques, balance symptoms, and timing. In MEDLINE (Pubmed), a preliminary search string was developed and then improved based on words in titles, abstracts, and index terms of highly relevant articles. It was then further improved through a PRESS [28] review with the assistance of a medical librarian. The improved search strategy is reported in S1 Appendix, and it will be adapted for each included database or information source. The reference lists of included studies and relevant systematic reviews will be examined to identify additional studies that meet the eligibility criteria. When relevant experimental details or data are missing, study authors will be contacted.

Studies must have been published in the last 10 years because there have been significant technological improvements in neuroimaging modalities that may make older literature obsolete to a current medical setting. Only studies published in English, or that can easily be translated into English by electronic means, will be included.

Elsevier (Embase), MEDLINE (PubMed), Google Scholar (first 200 hits), and SportDiscus (EBSCOhost) will be searched [29]. "Gray literature," which includes theses, dissertations, unpublished studies, ongoing research, and conference proceedings, will be included by searching ProQuest.

## Study selection

Saving results from searches, removal of duplicates, and management of the bibliography will be handled in EndNote version 21 (Clarivate Analytics, PA, USA), after which the review will be facilitated by Covidence software (Veritas Health Innovation, Melbourne, Australia). The de-duplicated list of studies from the literature search will be reviewed by two independent reviewers to evaluate the titles and abstracts of the articles relative to inclusion and exclusion criteria. Title and abstract screeners will first complete a pilot session. Articles that appear to meet inclusion criteria after title/abstract review will then be assessed by two trained reviewers

for eligibility by reading the full text of the article for inclusion and exclusion criteria. Reasons for excluding studies during full text review will be documented and reported. Any discrepancies found in the work of the two independent reviewers will be discussed with a third co-author to reach agreement on which studies should be included. Detailed results from the search and screening processes will be reported in a PRISMA flow diagram [19].

## Assessment of methodological quality

Studies included based on the full text review will be critically appraised by two independent reviewers, with disagreements resolved by a third. Critical appraisal will be based on standardized critical appraisal instruments for experimental and quasi-experimental (e.g., CASP [30]), observational studies (e.g., STROBE [31]), prognostic studies (QUIPS [21]), or by using instruments provided by the Joanna Briggs Institute (JBI).[20] The cited tools will assess the validity and specific sources of bias that reduce the quality of the study. Quality will be assessed at both the study level and the outcome level. Results from the study quality assessment will be used to exclude studies with scores less than 50% of the available scale. Results of the critical appraisal will be presented in a table and via narrative review.

## Data extraction

After included studies have been critically appraised, data extraction will be performed by two independent reviewers using the extraction tool in S2 Appendix and assisted by the R (R Core Team. 2023. r-project.org) module PDE (Stricker 2023. PDE: Extract Tables and Sentences from PDFs with User Interface v1.4.3) to minimize errors. Study-specific information will include overall design (e.g., cohort) and participant age, sex, most common cause of injury, time between injury and scan, and comorbidities. There will also be information about the neuroimaging modality and metrics pertaining to the modality, as well as the brain region(s) targeted. The method used to diagnose the balance problem and the length of follow-up will be noted. For each brain region and treatment group, the mean and standard deviation or odds ratio and risk ratio and associated standard error, along with sample size, will be recorded. If presented as continuous data, regression coefficients, correlation coefficients, or hazard ratios with associated standard errors and sample sizes will be collected. Disagreements will be resolved through consultation with a third reviewer.

## Data synthesis

The meta-analysis will be guided by Cochrane guidelines [32], and guidelines for prognostic study meta-analysis [21] will be incorporated as appropriate. Data from the studies will be pooled in a statistical meta-analysis using R software (R Core Team 2023. R Foundation for Statistical Computing) packages 'meta' [33] and 'metafor' [34]. Early phase prognostic studies can have significant heterogeneity in how they are reported, particularly with regard to the statistics employed and covariates. Adjusted and unadjusted coefficients will be analyzed separately, where covariates most likely to be included are loss of consciousness at the time of injury, age, and sex.

Effect sizes and 95% confidence intervals will be expressed as mean neuroimaging modality value differences or standardized mean differences. For analysis of symptom presentation, the difference will include the mean for the group with balance symptoms minus the mean for the group without balance symptoms. For symptom persistence, the difference will be the mean for the group with persistent balance symptoms minus the mean for the group without persistent balance symptoms. Standardized mean differences will be used for modalities which are measured on different scales for the same quantity and are not easily convertible. If there are

studies that also use odds ratios, hazard ratios, or regression coefficients, effects will be summarized using the C statistic or observed (O) compared to expected (O:E) statistics as discussed in [21].

There will be an overall pooled estimate for each modality, and the analysis will also estimate pooled estimates for brain regions within modalities. Studies with multiple modalities will be analyzed using a complex model which allows estimation of correlation coefficients between modalities to provide a more complete picture of the unique compared to the shared information provided by each modality. Brain regions will be analyzed individually when ten or more studies are found. For brain regions with fewer than ten studies, an analysis will be conducted on aggregated regions using a strategy of combining multiple regions in an individual study if required, or by splitting the comparator group and including sub-regions individually [35].

A random effects model based on the inverse variance method will be used, unless information is sparse enough to suggest the Mantel-Haenszel methods would be preferred [32]. Skewness of data suggesting a need for transformation or other methodologies will be considered relative to each modality. Heterogeneity will be assessed visually in a forest plot and also using the $I^2$ statistic and Cochrane's Q to determine appropriateness of the meta-analysis [32]. Meta-regressions associating outcomes with median time from injury to scan and age will also be conducted.

Results will be provided in a forest plots and scatterplots and summarized in the narrative. A sensitivity analysis will be performed to assess the impact of including studies with sample sizes an order of magnitude or more over the median sample size of included studies. Subgroup analysis is planned for mTBI diagnosis, age (pediatric, adult, older adult if available) and type of balance assessment. To assess the possibility of publication bias, funnel plots will be used for modalities with sufficient data, which will be accompanied by an Egger's test for funnel plot asymmetry [32].

## Assessing certainty in the findings

A summary of findings (SoF) tables will be prepared for the presentation and the persistence of balance symptoms based on the Grading in Recommendations, Assessment, Development, and Evaluation (GRADE) approach in GRADEpro GDT software (McMaster University and Evidence Prime 2024). For persistence of balance symptoms, the approach to GRADE will use phase of investigation as the starting point for assessing the quality of the evidence [36]. Two independent reviewers will construct the tables, with disagreements resolved by discussion. Each modality considered in the review will appear in the table. It is likely that evidence associated with individual brain regions will vary within modality. The SoF table will summarize the information for the brain region with the strongest evidence for association with balance problems for each modality based on number of studies included, quality of studies, and effect size from the meta-analysis. Modalities and variables in the summary of findings will include:

## Supporting information

**S1 Checklist. PRISMA-P (Preferred Reporting Items for Systematic review and Meta-Analysis Protocols) 2015 checklist: Recommended items to address in a systematic review protocol\*.**
(DOC)

**S1 Appendix. Search terms used in preliminary PubMed literature search – listed search terms used in this studies preliminary review of the literature and their results.**
(DOCX)

**S2 Appendix. List of variables for data extraction - variables used for the various components of this review's search and analysis and their associated units used for analysis.** (DOCX)

## Acknowledgments

We appreciate the contributions of Catie Chung and Kelly Mecham, who have made this protocol possible by bringing JBI training and materials to our institution. Medical Librarians Megan De Armond and Katie Hoskins help develop and review the search strategy.

## Author Contributions

**Conceptualization:** Todd Boscarello, Robby Boparai, Nathan Samson, Alan Rodriguez, Cheryl Vanier.

**Data curation:** Cheryl Vanier.

**Formal analysis:** Cheryl Vanier.

**Methodology:** Robby Boparai, Cheryl Vanier.

**Project administration:** Alan Rodriguez, Thomas Knoblauch, Cheryl Vanier, Travis Snyder.

**Resources:** Alan Rodriguez, Thomas Knoblauch, Cheryl Vanier, Travis Snyder.

**Software:** Alan Rodriguez, Thomas Knoblauch, Cheryl Vanier.

**Supervision:** Todd Boscarello, Alan Rodriguez, Thomas Knoblauch, Travis Snyder.

**Validation:** Alan Rodriguez.

**Writing – original draft:** Todd Boscarello, Robby Boparai, Nathan Samson, Alan Rodriguez, Cheryl Vanier.

**Writing – review & editing:** Todd Boscarello, Robby Boparai, Alan Rodriguez, Cheryl Vanier.

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
