## [Decision Letter · Decision Letter 0]

8 Sep 2024

PONE-D-24-26492

Neuroimaging Findings and Balance Problems after Mild Traumatic Brain Injury: A Systematic Review Protocol

PLOS ONE

Dear Dr. Boscarello,

Thank you for submitting your manuscript to PLOS ONE. After careful consideration, we have decided that your manuscript does not meet our criteria for publication and must therefore be rejected.

Specifically:

The reviewers have noted that the current version of the paper lacks sufficient data to support the findings. Please ensure that the objectives of your study are thoroughly addressed and make every effort to respond to the reviewers' comments in your next effort.

I am sorry that we cannot be more positive on this occasion, but hope that you appreciate the reasons for this decision.

Kind regards,

Jose Gerardo Tamez-Peña, PhD

Academic Editor

PLOS ONE

Reviewers' comments:

Reviewer's Responses to Questions

**Comments to the Author**

1. Does the manuscript provide a valid rationale for the proposed study, with clearly identified and justified research questions?

Reviewer #1: Yes

Reviewer #2: Yes

2. Is the protocol technically sound and planned in a manner that will lead to a meaningful outcome and allow testing the stated hypotheses?

Reviewer #1: Partly

Reviewer #2: Yes

3. Is the methodology feasible and described in sufficient detail to allow the work to be replicable?

Reviewer #1: Yes

Reviewer #2: Yes

4. Have the authors described where all data underlying the findings will be made available when the study is complete?

Reviewer #1: Yes

Reviewer #2: No

5. Is the manuscript presented in an intelligible fashion and written in standard English?

Reviewer #1: Yes

Reviewer #2: Yes

6. Review Comments to the Author

You may also provide optional suggestions and comments to authors that they might find helpful in planning their study.

Reviewer #1: The authors present a protocol to perform a meta-analysis study regarding neuroimaging findings that correlate with loss or deficit in balance within people that suffered a mild traumatic brain injury. Neuroimaging modalities to be included are structural MRI, metrics from diffusion tensor imaging and functional connectivity assessed using functional MRI. In addition, they will include results from EEG and MEG.

The protocol aims at finding correlations between neuroimaging metrics and loss in balance of patients that suffered a mild traumatic brain injuries. It establishes a good inclusion and exclusion criteria as well as data analysis methodology. I believe it may yield successful results, however, there are a few observations:

- Authors do not mention how will variability of MRI acquisition parameters will be handled. This is, MR system vendor, pulse sequences, magnetic field strength, dwi parameters such as number of diffusion directions, voxel size, preprocessing pipeline as well as fmri acquisition parameters. I believe it is important that authors include an strategy to deal with such variability sources. In MRI literature, such parameters are related as sources of variability and may lead to confounding results.

- It is not entirely clear whether the authors will analyze results reported directly in the publications or if they will request for the raw data and re-analyze it.

Reviewer #2: This manuscript has no data, only a description of a proposed study. The reasons for submitting such a study are unclear.

7. PLOS authors have the option to publish the peer review history of their article (what does this mean?). If published, this will include your full peer review and any attached files.

Reviewer #1: No

Reviewer #2: No

- - - - -

---

## [Author Response · Author response to Decision Letter 0]

22 Sep 2024

The previous rejection was contested on the basis of misunderstanding the significance of a protocol. The primary reason for rejection was lack of results, which protocols inherently do not contain. This rejection was appealed, after which we were advised to resubmit.

---

## [Decision Letter · Decision Letter 1]

28 Nov 2024

PONE-D-24-26492R1Neuroimaging Findings and Balance Problems after Mild Traumatic Brain Injury: A Systematic Review ProtocolPLOS ONE

Dear Dr. Boscarello,

Thank you for submitting your manuscript to PLOS ONE. After careful consideration, we feel that it has merit but does not fully meet PLOS ONE’s publication criteria as it currently stands. Therefore, we invite you to submit a revised version of the manuscript that addresses the points raised during the review process.

We look forward to receiving your revised manuscript.

Kind regards,

Maryam Bemanalizadeh

Academic Editor

PLOS ONE

Journal Requirements:

Additional Editor Comments (if provided):

Reviewers' comments:

Reviewer's Responses to Questions

**Comments to the Author**

1. Does the manuscript provide a valid rationale for the proposed study, with clearly identified and justified research questions?

Reviewer #3: Yes

Reviewer #4: Yes

Reviewer #5: Yes

2. Is the protocol technically sound and planned in a manner that will lead to a meaningful outcome and allow testing the stated hypotheses?

Reviewer #3: Yes

Reviewer #4: Yes

Reviewer #5: Yes

3. Is the methodology feasible and described in sufficient detail to allow the work to be replicable?

Reviewer #3: Yes

Reviewer #4: Yes

Reviewer #5: Yes

4. Have the authors described where all data underlying the findings will be made available when the study is complete?

Reviewer #3: No

Reviewer #4: Yes

Reviewer #5: No

5. Is the manuscript presented in an intelligible fashion and written in standard English?

Reviewer #3: Yes

Reviewer #4: Yes

Reviewer #5: Yes

6. Review Comments to the Author

You may also provide optional suggestions and comments to authors that they might find helpful in planning their study.

Reviewer #3: A study protocol is an important document that specifies the research plan and should be written in detail. I think the researchers fully describe the research question and justify the need for the study.

Reviewer #4: Dear Authors, thank you for your contribution.

I would like to know the purpose of publishing a Systematic Review and Meta-Analysis protocol without having done it.

Reviewer #5: The protocol is sound. The only obvious issue is that there is no statement identifying where the data for the study will be made available. This wouldn't be an issue until publication of the article if the protocol were not being independently published first. In that case, question 4 above (Have the authors described where all data underlying the findings will be made available when the study is complete?) must be answered no. Adding this statement should be very easy.

I do have clinical questions about their methodology, such as the decision to limit analysis to mild TBI, but that should not hinder the progress of their protocol. Beyond that, the study protocol largely meets the expectations of the journal and should be allowed to proceed.

7. PLOS authors have the option to publish the peer review history of their article (what does this mean?). If published, this will include your full peer review and any attached files.

Reviewer #3: No

Reviewer #4: No

Reviewer #5: No

---

## [Author Response · Author response to Decision Letter 1]

29 Nov 2024

Below are the comments and questions from the various reviewers. My responses are in the outline as "a."

4. Reviewer #4: Dear Authors, thank you for your contribution.

I would like to know the purpose of publishing a Systematic Review and Meta-Analysis protocol without having done it.

a. This is an excellent question. The purpose of publishing a systematic review or meta analysis protocol is to ensure transparency, minimize bias, and perhaps most importantly add to the literature regarding how to perform a meta analysis with difficult to compare modalities. For example, this study is comparing imaging modalities, and our protocol outlines how to compare the variable of the various modalities. This may prove useful for others who desired to perform studies looking at different pathology comparing findings from different imaging modalities. 

5. Reviewer #5: The protocol is sound. The only obvious issue is that there is no statement identifying where the data for the study will be made available. This wouldn't be an issue until publication of the article if the protocol were not being independently published first. In that case, question 4 above (Have the authors described where all data underlying the findings will be made available when the study is complete?) must be answered no. Adding this statement should be very easy. I do have clinical questions about their methodology, such as the decision to limit analysis to mild TBI, but that should not hinder the progress of their protocol. Beyond that, the study protocol largely meets the expectations of the journal and should be allowed to proceed.

a. Yes, the data is currently being collected, and thus this question likely does not apply to the publication of a protocol. Please advise if it would be prudent to add an addendum when the data is available. We would be more than happy to comply.

b. Regarding your question on why we are just limiting our study to mTBI, that is because mTBI and TBI are differentiated historically by imaging findings. TBI has significant imaging findings, while mTBI is generally characterized by no imaging findings. Of course, this study is examining the truth to this assumption, finding that long term complications after mTBI do indeed have subtle imaging findings in the long term. We also chose just mTBI in order to keep the scope narrow enough to be manageable for data acquisition and management purposes.

---

## [Editor Report · Decision Letter 2]

16 Dec 2024

Neuroimaging Findings and Balance Problems after Mild Traumatic Brain Injury: A Systematic Review Protocol

PONE-D-24-26492R2

Dear Dr. Boscarello,

We’re pleased to inform you that your manuscript has been judged scientifically suitable for publication and will be formally accepted for publication once it meets all outstanding technical requirements.

Kind regards,

Academic Editor

PLOS ONE
---

## [Editor Report · Acceptance letter]

20 Dec 2024

PONE-D-24-26492R2 

PLOS ONE

Dear Dr. Boscarello, 

I'm pleased to inform you that your manuscript has been deemed suitable for publication in PLOS ONE. Congratulations! Your manuscript is now being handed over to our production team.

Kind regards, 

on behalf of

Dr. Maryam Bemanalizadeh 

Academic Editor

PLOS ONE